# SCGB1C1 Plays a Critical Role in Suppression of Allergic Airway Inflammation through the Induction of Regulatory T Cell Expansion

**DOI:** 10.3390/ijms25116282

**Published:** 2024-06-06

**Authors:** Sung-Dong Kim, Shin-Ae Kang, Sue-Jean Mun, Hak-Sun Yu, Hwan-Jung Roh, Kyu-Sup Cho

**Affiliations:** 1Department of Otorhinolaryngology, Pusan National University Hospital, Busan 49241, Republic of Korea; applekims@hanmail.net; 2Department of Parasitology and Tropical Medicine, Pusan National University School of Medicine, Yangsan 50612, Republic of Korea; f2s4u@pusan.ac.kr (S.-A.K.); hsyu@pusan.ac.kr (H.-S.Y.); 3Department of Otorhinolaryngology, Pusan National University Yangsan Hospital, Yangsan 50612, Republic of Korea; baskie23@naver.com (S.-J.M.); rohhj@pusan.ac.kr (H.-J.R.)

**Keywords:** mesenchymal stromal cells, adipose stem cells, extracellular vesicles, immunosuppression, secretoglobin family 1C member 1

## Abstract

The nanosized vesicles secreted from various cell types into the surrounding extracellular space are called extracellular vesicles (EVs). Although mesenchymal stem cell-derived EVs are known to have immunomodulatory effects in asthmatic mice, the role of identified pulmonary genes in the suppression of allergic airway inflammation remains to be elucidated. Moreover, the major genes responsible for immune regulation in allergic airway diseases have not been well documented. This study aims to evaluate the immunomodulatory effects of secretoglobin family 1C member 1 (SCGB1C1) on asthmatic mouse models. C57BL/6 mice were sensitized to ovalbumin (OVA) using intraperitoneal injection and were intranasally challenged with OVA. To evaluate the effect of SCGB1C1 on allergic airway inflammation, 5 μg/50 μL of SCGB1C1 was administrated intranasally before an OVA challenge. We evaluated airway hyperresponsiveness (AHR), total inflammatory cells, eosinophils in the bronchoalveolar lavage fluid (BALF), lung histology, serum immunoglobulin (Ig), the cytokine profiles of BALF and lung-draining lymph nodes (LLN), and the T cell populations in LLNs. The intranasal administration of SCGB1C1 significantly inhibited AHR, the presence of eosinophils in BALF, eosinophilic inflammation, goblet cell hyperplasia in the lung, and serum total and allergen-specific IgE. SCGB1C1 treatment significantly decreased the expression of interleukin (IL)-5 in the BALF and IL-4 in the LLN, but significantly increased the expression of IL-10 and transforming growth factor (TGF)-β in the BALF. Furthermore, SCGB1C1 treatment notably increased the populations of CD4^+^CD25^+^Foxp3^+^ regulatory T cells (Tregs) in asthmatic mice. The intranasal administration of SCGB1C1 provides a significant reduction in allergic airway inflammation and improvement of lung function through the induction of Treg expansion. Therefore, SCGB1C1 may be the major regulator responsible for suppressing allergic airway inflammation.

## 1. Introduction

Bronchial asthma is a chronic inflammatory airway disease, with the key features of persistent airway inflammation, airway hyperresponsiveness (AHR), and airway remodeling [1]. Insufficient regulatory T cell (Treg) suppression, leading to disproportionate Th2 cell activation, is acknowledged as being a pivotal factor in the pathogenesis of allergic airway inflammation [2,3,4]. Furthermore, airway remodeling has been reported to be important in pathological pathways of asthma, characterized by irreversible AHR and airway obstruction [5]. Several studies have reported that mesenchymal stem cells (MSCs), including those derived from adipose tissue (ASCs), can ameliorate allergic airway inflammation by upregulating Tregs and increasing the levels of soluble factors, such as prostaglandin E2 (PGE2) and transforming growth factor-β (TGF-β) [6,7,8,9].

The MSC secretome or MSC-derived extracellular vesicles (EVs) were as effective as MSCs themselves in improving allergic airway diseases [10,11,12,13,14,15]. The EVs of ASCs ameliorated Th2-mediated inflammation through the activation of dendritic cells and the induction of M2 macrophage polarization [16,17,18]. The immunomodulatory effects of MSCs-derived EVs in allergic airway inflammation could be mediated by the upregulation of Tregs and the increased expression of pulmonary genes, such as paraoxonase-1 (PON1), brain-expressed X-linked 2 (Bex2), insulin-like growth factor binding protein 6 (Igfbp6), and secretoglobin family 1C member 1 (SCGB1C1) [19]. However, the pivotal pulmonary genes underlying the immunomodulatory effects of MSC-derived EVs in allergic airway diseases remain poorly understood, and the specific roles of these genes in the suppression of allergic airway inflammation by MSC-derived EVs have yet to be elucidated.

In the study, we evaluated the effects of SCGB1C1 on AHR; eosinophilic inflammation; serum immunoglobulin; Th1-, Th2- and Treg-related cytokines; and the modulation of Tregs in an ovalbumin (OVA)-induced asthmatic mouse model.

## 2. Results 

### 2.1. AHR and Inflammatory Cells in BALF

The Penh values of four groups were increased with the escalating methacholine concentration. The Penh values in asthmatic mice at 50 mg/mL were significantly higher than those in the PBS and SCGB1C1 group (all *p* < 0.001). The intranasal administration of SCGB1C1 significantly decreased AHR in asthmatic mice (*p* = 0.049) (Figure 1A). The total inflammatory cell and eosinophil counts were remarkably increased in the BALF of the OVA group compared to the PBS group (all *p* < 0.001). However, intranasal treatment with SCGB1C1 markedly lowered the numbers of eosinophils in asthmatic mice (*p* = 0.049) (Figure 1B).

### 2.2. Lung Histology and Inflammation Score 

We found no obvious infiltration of inflammatory cells in the PBS or SCGB1C1 group. Asthmatic mice showed significant eosinophil infiltration around the peribronchiolar and perivascular areas. Goblet cell hyperplasia, indicated by the increased number and size of PAS-stained goblet cells, was found in the respiratory epithelium of the OVA group. However, treatment with SCGB1C1 remarkably decreased infiltration of inflammatory cells and goblet cell hyperplasia (Figure 2A). Furthermore, the peribronchiolar and perivascular inflammation score was significantly decreased in the SCGB1C1-treated group compared to the OVA group (*p* < 0.001 and *p* = 0.012, respectively) (Figure 2B). 

### 2.3. Serum Total and OVA-Specific IgE, IgG1, and IgG2a Levels

The total and OVA-specific IgE (*p* = 0.035 and *p* = 0.006, respectively) and IgG1 (*p* = 0.004 and *p* = 0.011, respectively) levels were significantly increased in the OVA group compared to the PBS group. However, the intranasal administration of SCGB1C1 significantly lowered the total IgE (*p* = 0.037) and OVA-specific IgE (*p* = 0.009) in asthmatic mice. There were no significant changes in the total and OVA-specific IgG1 and IgG2a levels (Figure 3). 

### 2.4. Expression of Cytokines in the BALF and LLNs

The level of IL-5 in the BALF was significantly higher in the OVA group than in the PBS group (*p <* 0.001). However, intranasal SCGB1C1 treatment significantly decreased IL-5 in the BALF (*p* = 0.039) and IL-4 in the LLN (*p* = 0.040) of asthmatic mice. In contrast, intranasal SCGB1C1 treatment remarkably increased IL-10 and TGF-β in the BALF of asthmatic mice (*p* = 0.011 and *p* = 0.026, respectively) (Figure 4).

### 2.5. T Cell Population in the LLNs

The population of CD4^+^CD25^+^Foxp3^+^ T cells in LLNs was markedly increased by the intranasal administration of SCGB1C1 in asthmatic mice. In the OVA+SCGB1C1 group, CD4^+^CD25^+^Foxp3^+^ T cells were significantly increased compared to the OVA group (*p* = 0.042) (Figure 5).

## 3. Discussion

EVs are spherical bi-layered proteolipids secreted from almost all live cells into extracellular spaces. Depending on their origin, size, and biogenesis, EVs can be subdivided into apoptotic bodies, microvesicles, and exosomes [20]. MSC-derived EVs, which are the main paracrine effectors of stem cells, play a crucial role in intracellular communication by transferring important biomolecules [21]. MSCs-derived EVs have been reported to be promising candidates for the treatment of allergic airway diseases due to their immunomodulatory properties [10,22]. Several studies have shown that ASC-derived EV treatment significantly improved allergic airway inflammation through the suppression of Th2 cytokine production and induction of Treg expansion [15,18]. Furthermore, ASC-derived EVs have been shown to reduce AHR and improve lung function in asthmatic mice [15]. EVs exert their biological functions by transporting various cellular components, such as proteins, mRNAs, and microRNAs, and DNA, to recipient cells [10].

Recently, we performed screening and functional pathway analysis of the pulmonary genes associated with the suppression of allergic airway inflammation by ASC-derived EVs. The gene expression levels of PON1 and SCGB1C1 decreased in the lung tissue of asthmatic mice, but PON1 and SCGB1C1 expression significantly increased after treatment with ASC-derived EVs [19]. Regarding the immune mechanisms by which ASC-derived EVs regulate allergic airway diseases, the microRNAs and pulmonary genes released by ASC-derived EVs induce the expansion of Tregs, which leads to a decrease in the release of allergy-specific Th2 cytokines, eosinophil infiltration, and allergy-specific IgG1 and IgE production in asthmatic mice [19,23]. These findings suggest that SCGB1C1 may play a critical role in the immune suppression mechanisms of ASC-derived EVs in allergic airway diseases. Considering the evidence suggesting the immunomodulatory effects of ASC-derived EVs mediated by SCGB1C1, the plausible roles of SCGB1C1 in an asthmatic mouse model were investigated in this study [24].

Secretoglobins (SCGBs) are highly expressed in the body fluids of the uterus, prostate salivary glands, lacrimal glands, lungs, and in other tissues [25]. The human airway serves as the primary site for mRNA expression of SCGB family members, and SCGBs are predominantly secreted proteins found exclusively in mammals [26,27]. Although the precise biological activities of most SCGBs remain incompletely understood, this family has an important role in the immunoregulatory and anti-inflammatory process of airway diseases [28]. SCGB1A1 is involved in the pathophysiology of allergic diseases. Serum SCGB1A1 levels were found to be lower in patients with asthma and allergic rhinitis compared to healthy controls. Allergen-specific immunotherapy increased the expression of SCGB1A1-suppressing osteopontin, a cytokine with Th2-promoting functions in the upper and lower airways [29,30]. SCGB1C1 is notably expressed in the mucosa of the human respiratory tract. Furthermore, SCGB1C1 is downregulated by interferon (IFN)-γ and upregulated by IL-4 and IL-13 [25,26,27]. The SCGB1C1protein can modulate the inflammation process by binding the steroid ligand, resulting in the prevention of nasal polyp development [31]. In our recent study, the SCGB1C1 mRNA levels were lower in an OVA-induced asthmatic mice, and these were upregulated following treatment with ASC-derived EVs [19].

The present study demonstrated that intranasal administration of SCGB1C1 to asthmatic mice provides a significant reduction in allergic airway inflammation and a marked alleviation of AHR. SCGB1C1 treatment significantly decreased the eosinophils in the BALF and improved eosinophilic lung inflammation. Together with the increased expression of SCGB1C1 in lung tissue by ASC-derived EVs, these findings strongly suggest that SCGB1C1 suppressed eosinophil recruitment to the lung and BALF and improved AHR in asthmatic mice, which is consistent with the results of previous studies using ASC supernatants or ASC-derived EVs [12,15]. Furthermore, SCGB1C1 treatment significantly lowered the total and OVA-specific IgE levels. The levels of IL-5 in the BALF and IL-4 in the LLNs were significantly decreased, whereas IL-10 and TGF-β levels were significantly increased in the BALF following intranasal SCGB1C1 treatment. The population of Tregs in SCGB1C1-treated asthmatic mice was significantly higher than that in untreated mice, which was similar to the results of previous studies indicating that SCGB1C1 preferentially activates CD4^+^CD25^+^Foxp3^+^ T cells, which is the main mechanism of immunomodulatory effects of SCGB1C1. Together with the increased expression of SCGB1C1 in lung tissue by ASC-derived EVs, these findings strongly indicate that SCGB1C1 is involved in the suppression of allergic airway inflammation, mediated by ASC-derived EVs. 

Our study has some limitations. To clarify these findings, it is necessary to investigate the immunomodulatory effects of SCGB1C1 on allergic airway inflammation in patients with asthma and allergic rhinitis. Future studies will address the exact receptors and signaling cascade responsible for the immunomodulatory mechanisms of SCGB1C1 in allergic airway diseases. 

## 4. Materials and Methods

### 4.1. Animals

Six-week-old female C57BL/6 mice were purchased from Samtako Co. (Osan, Republic of Korea) and housed in a specific pathogen-free animal facility throughout the duration of the experiments. The Institutional Animal Care and Use Committee of the Pusan National University School of Medicine approved the animal study protocol (Approval No. PNU-2016-1109).

### 4.2. Mouse Model of Allergic Airway Inflammation

A mouse model of allergic airway inflammation was established, following a protocol with minor modifications from previous studies [15,32]. C57BL/6 mice were sensitized via intraperitoneal injection with 75 μg of OVA (Sigma-Aldrich, St. Louis, MO, USA) and 2 mg of aluminum hydroxide (Sigma-Aldrich) into 200 μL of phosphate-buffered saline (PBS) on days 0, 1, 7, and 8. Subsequently, the mice were challenged intranasally with 50 μg of OVA in 50 μL PBS on days 14, 15, 21, and 22 after the initial sensitization (Figure 6A). 

### 4.3. Intranasal Administration of SCGB1C1

SCGB1C1 peptides were chemically synthesized from Biostem (Ansan, Republic of Korea). The amino acid sequence of a SCGB1C1 protein is EDNDEFF MDFLQTLLVG TPEELYEGTL GKYNVNEDAK AAMTELKSCI DGLQPMHKAE LVKLLVQVLG SQDGA. To evaluate the effect of SCGB1C1 on allergic airway disease, the intranasal administration of SCGB1C1 (5 μg/50 μL) was performed on days 12, 13, 19, and 20. The mice were divided into four groups, each consisting of four mice: (a) a PBS group sensitized, pretreated, and challenged with PBS; (b) a SCGB1C1 group sensitized with OVA, pretreated with SCGB1C1, and then challenged with PBS; (c) OVA group sensitized with OVA, pretreated with PBS, and then challenged with OVA; and (d) OVA+SCGB1C1 group sensitized with OVA, intranasally pretreated with SCGB1C1, and then challenged with OVA (Figure 6B). 

### 4.4. Measurement of Airway AHR

As outlined in a prior study, AHR was evaluated in conscious and unrestrained mice using noninvasive whole-body plethysmography (Allmedicus, Seoul, Republic of Korea) 24 h from the last challenge [15,32]. Mice were exposed to increasing concentrations of aerosolized methacholine (0, 12.5, 25, and 50 mg/mL) for 10 min in a plethysmography chamber. The enhanced pause (Penh) was calculated based on the average pressure generated during both inspiration and expiration during the time of each phase. Penh values were automatically recorded at 3 min intervals and averaged. 

### 4.5. Differential Cell Counting in Bronchoalveolar Lavage Fluid (BALF)

The BLAF was collected following the procedures outlined in previous studies [12,15,32,33]. After mice were anesthetized, the tracheas were exposed and cut just below the larynx. A flexible polyurethane tube was attached to a blunt 24-gauge needle, with a 0.4-mm outer diameter and length of 4 cm (Boin Medical Co., Seoul, Republic of Korea); 800 μL of cold PBS was inserted into the trachea. Some 800 μL of cold PBS was slowly drawn into the trachea using a 1 mL syringe. Then, the syringe was pulled to recover the PBS immediately and gently. The PBS was placed back into the trachea, and the syringe was pulled again to recover the BALF. The amount of BALF recovered varied by group, but was approximately 400–600 μL. BALF samples were centrifuged at 1500 rpm and 4 °C for 5 min, and the supernatants were immediately frozen at −70 °C. The total cell counts were measured after resuspending the cell pellet and washing it in PBS. BALF cell smears were obtained via staining with a cytospin device and Diff-Quik solution (Sysmex Co., Kobe, Japan), and differential cells were identified according to established morphological criteria. Differential leukocyte counts were obtained by evaluating a minimum of 500 cells per slide. 

### 4.6. Lung Histology and Inflammation Scoring

Following lavage, the lung tissue was removed, fixed in 10% neutral formalin for 36 h, and then embedded in paraffin. Hematoxylin and eosin (H&E) and periodic acid Schiff (PAS) staining were performed to identify eosinophils and mucin-secreting cells, respectively. Lung inflammation was determined by the peribronchiolar and perivascular inflammation levels, rated on a subjective scale from 0 to 4, as described in previous studies [15,20,21,22]. The thin sections of the embedding tissues were stained with hematoxylin and eosin (H&E) and periodic acid Schiff (PAS) for the identification of eosinophils and mucin-secreting cells, respectively. The index of lung inflammation was determined by the level of peribronchial and perivascular inflammation (0, normal; 1, ≤3 cells thick; 2, 4–10 cells thick; and 3, ≥10 cells thick) and the overall extent of inflammation (0, normal; 1, <25% of the sample; 2, 25–50% of the sample; 3, 51–75% of the sample; 4, ≥75% of the sample) [12,15,32,33]. The score was calculated by multiplying the severity by the extent [12,15,32,33]. 

### 4.7. Measurement of Serum Immunoglobulin

Mice serum was collected through a cardiac puncture at 48 h following the final OVA challenge. Total and OVA-specific immunoglobulins (IgE, IgG1, IgG2a) were quantified using an enzyme-linked immunosorbent assay (ELISA) according to the manufacturer’s instructions (R&D Systems, Minneapolis, MN, USA). The quantification of total and OVA-specific immunoglobulins (IgE, IgG1, and IgG2a) was performed using an enzyme-linked immunosorbent assay (ELISA), following the manufacturer’s instructions (R&D Systems, Minneapolis, MN, USA). The absorbance at 450 nm was measured using an ELISA plate reader (Molecular Devices, Sunnyvale, CA, USA). The absorbance of each well at 450 nm was captured using an ELISA plate reader (Molecular Devices, Sunnyvale, CA, USA).

### 4.8. Expression of Cytokines in the Lung-Draining Lymph Nodes and BALF

Lung-draining lymph nodes (LLNs) were obtained between the trachea and both lung lobes in asthmatic mice. Lymphocytes were isolated from LLNs and cultured in 48-well plates, coated with 0.5 μg/mL of CD3 antibody (BD Pharmingen™, BD Bioscience, San Jose, CA, USA) at a concentration of 10^6^ cells/mL in RPMI 1640 with 10% fetal bovine serum (FBS). All cells were maintained for 72 h at 37 °C with 5% CO_2_. The concentrations of interleukin (IL)-4, IL-5, IL-10, and TGF-β in the BALF and in the stimulated LLNs supernatants were examined using ELISA kits following the manufacturer’s recommendations (eBioscience, San Diego, CA, USA). The absorbance was measured at 450 nm on an ELISA plate reader (Molecular Devices, Sunnyvale, CA, USA).

### 4.9. Phenotypic Analysis of CD4^+^CD25^+^FOXP3^+^ Tregs

LLNs cells were washed with FACS buffer and cell pellets were labelled with a mouse Treg Staining kit (eBioscience) according to the manufacturer’s instructions. Briefly, cells were incubated with anti-CD4-FITC (0.5 mg/mL) and anti-CD25-APC antibodies (0.2 mg/mL) for 30 min at room temperature in dark. Following fixation with a Cytofix/Cytoperm kit (eBioscience) and permeabilization, cells were incubated with anti-Foxp3-PE-cy7 antibodies (0.2 mg/mL) (eBioscience). After the washing step, the supernatant was discarded, and fluorescence was measured using a BD FACSCanto™ II cytometer (BD Bioscience).

### 4.10. Statistical Analysis

All experiments were performed at least three times. Data were expressed as means ± standard deviations. Statistical significance was determined by a Student’s *t* test using GraphPad Prism 5.0 software (GraphPad Software Inc., La Jolla, CA, USA). A value of *p* < 0.05 was considered significant. 

## 5. Conclusions

The induction of SCGB1C1 by ASC-derived EVs significantly reduced allergic airway inflammation and improved lung function through the induction of Treg expansion in asthmatic mice. Therefore, SCGB1C1 may be a key regulator responsible for suppressing allergic airway inflammation through ASC-derived EVs. 

## Figures and Tables

**Figure 1 ijms-25-06282-f001:**
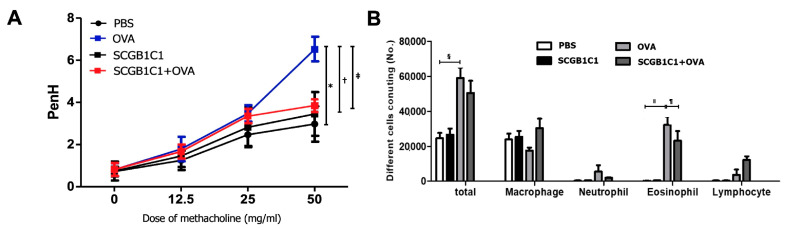
Effect of secretoglobin family 1C member 1 (SCGB1C1) on airway hyperresponsiveness (AHR) and inflammatory cells in the bronchoalveolar lavage fluid (BALF). (**A**) After methacholine challenge, AHR level was decreased significantly in the OVA+SCGB1C1 group compared to that in the OVA group. (**B**) Eosinophil counts in BALF were significantly lowered in the OVA+SCGB1C1 group compared to those in the OVA group. Data are expressed as the mean ± SD of four independent experiments, each performed in triplicate. *, †, §, ǁ *p* < 0.001, ‡ *p* = 0.049, ¶ *p* = 0.049.

**Figure 2 ijms-25-06282-f002:**
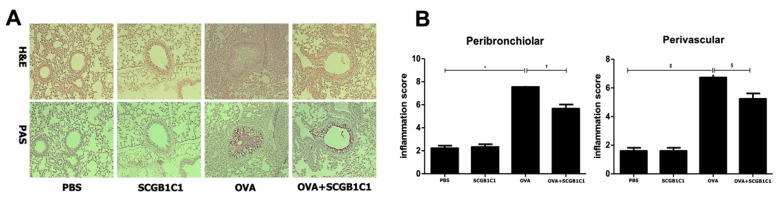
Effects of secretoglobin family 1C member 1 (SCGB1C1) on lung histology and inflammation score. (**A**) Infiltration of eosinophils and PAS-stained goblet cells showed a greater decrease in the OVA+SCGB1C1 group than in the OVA group (H&E, PAS ×200). (**B**) The inflammation score decreased significantly in the OVA+SCGB1C1 group compared to the OVA group around peribronchiolar and perivascular areas. Data are expressed as the mean ± SD of four independent experiments, each performed in triplicate. *, †, ‡ *p* < 0.001, § *p* = 0.012.

**Figure 3 ijms-25-06282-f003:**
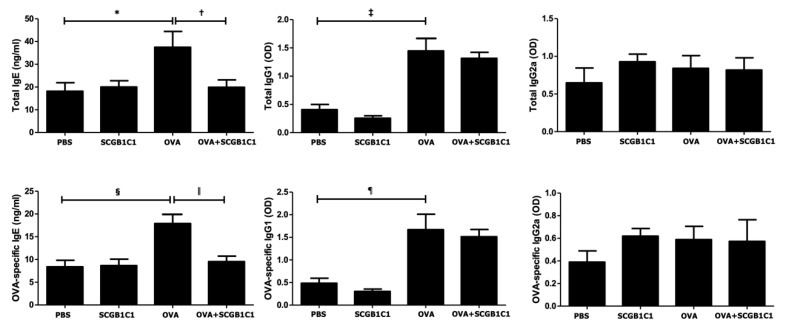
Effects of secretoglobin family 1C member 1 (SCGB1C1) on serum immunoglobulin levels. Total and OVA-specific IgE and IgG1 levels were significantly increased in the OVA group compared to those in the PBS group. Intranasal administration of SCGB1C1 significantly reduced the total and OVA-specific IgE in asthmatic mice. Data are expressed as the mean ± SD of four independent experiments, each performed in triplicate. * *p* = 0.035, † *p* = 0.037, ‡ *p* = 0.004, § *p* = 0.006, ǁ *p* = 0.009, ¶ *p* = 0.011.

**Figure 4 ijms-25-06282-f004:**
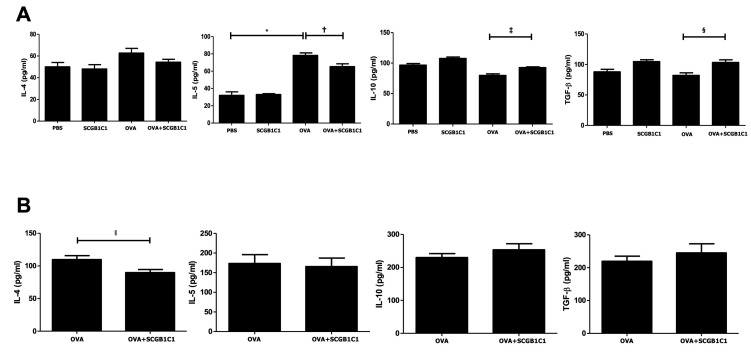
Effects of secretoglobin family 1C member 1 (SCGB1C1) on cytokine levels of the bronchoalveolar lavage fluid (BALF) (**A**) and lung-draining lymph nodes (LLNs) (**B**). IL-5 level was significantly increased in the BALF of the OVA group compared to that in the PBS group. Intranasal SCGB1C1 treatment remarkably reduced the levels of IL-5 but increased the levels of IL-10 and TGF-β in the BALF. Intranasal SCGB1C1 treatment significantly decreased the levels of IL-4 in the LLNs. Data are expressed as the mean ± SD of four independent experiments, each performed in triplicate. * *p* < 0.001, † *p* = 0.039, ‡ *p* = 0.011, § *p* = 0.026, ǁ *p* = 0.040.

**Figure 5 ijms-25-06282-f005:**
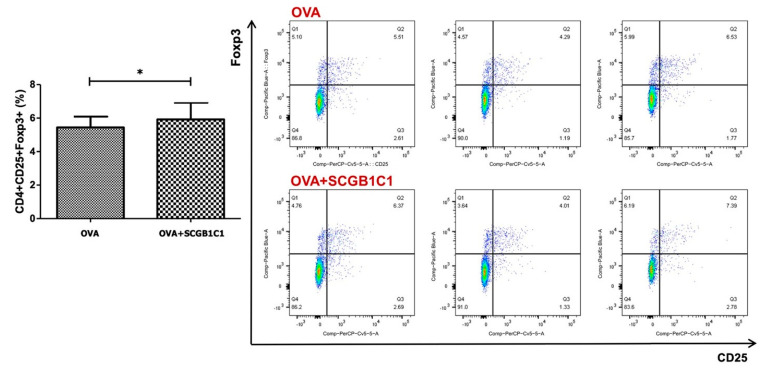
Effect of secretoglobin family 1C member 1 (SCGB1C1) on T cell populations in the lung-draining lymph nodes. The populations of CD4^+^CD25^+^Foxp3^+^ T cells were significantly higher in the OVA+SCGB1C1 group than OVA group. Data are expressed as the mean ± SD of four independent experiments, each performed in triplicate. * *p* = 0.042.

**Figure 6 ijms-25-06282-f006:**
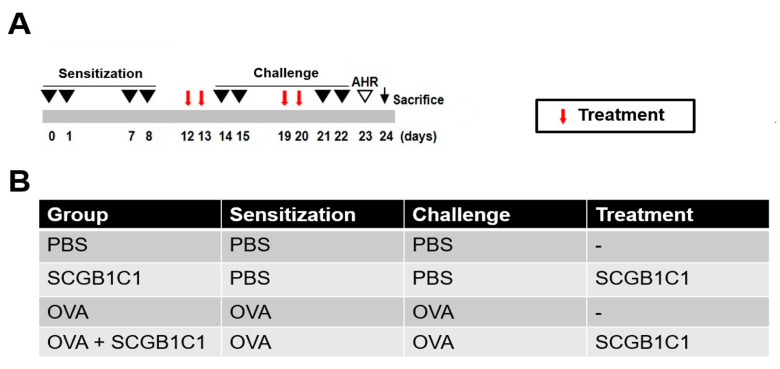
The experimental protocol. (**A**) The mice were sensitized on days 0, 1, 7, and 8 (black triangle) by intraperitoneal injection with 75 μg of ovalbumin (OVA) and 2 mg of aluminum hydroxide in 200 μL of PBS. Subsequently, the mice were challenged intranasally with 50 μg OVA in 50 μL of PBS on days 14, 15, 21, and 22 (black triangle) with OVA. On days 12, 13, 19, and 20, 5 μg/50 μL of the secretoglobin family 1C member 1 (SCGB1C1) was administered intranasally. (**B**) The mice were divided into four groups according to the different sensitization, challenge, and treatment methods. Four mice were used in each group.

## Data Availability

The original contributions presented in the study are included in the article, further inquiries can be directed to the corresponding author.

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
