# Peer review of "SCGB1C1 Plays a Critical Role in Suppression of Allergic Airway Inflammation through the Induction of Regulatory T Cell Expansion"

_ijms, 2024, doi:10.3390/ijms25116282_

Round 1

Reviewer 1 Report

Comments and Suggestions for Authors

After careful review, I have several major comments:

1.     “Abstract”: The authors should provide definitions for any abbreviated terms at first use. For terms used once or twice, abbreviation may not be needed. In addition, the findings from the study were not solid enough to draw the conclusions.

2.     “2.3. Intranasal administration of SCGB1C1”: Is there any relations between SCGB1C1 and adipose stem cells-derived extracellular vesicles? If no, the title of the manuscript may not be appropriate.

3.     Figure 1: It is important to mention the number of mice for each group.

4.     In Figure 1A, “Sensitization”, “Challenge”, and “Treatment” may be more appropriate to illustrate the timepoints of intervention. The agents used for sensitization, challenge, and treatment can be describe as the footers of the table in Figure 1B.

5.     “2.5. Differential cell counting in bronchoalveolar lavage fluid (BALF)”: A brief description of the procedure to collect BLAF is needed. The flush volume, the collection volume, and the times of flushes have great impact on the results.

6.     In all figures of the Results, the number of mice for each group should be addressed. The author mentioned that “Data are expressed as the mean ± SD of four independent experiments, each performed in triplicate”. Were the samples of the four independent experiments derived from four individuals? For a bar plot, the average datum of an individual can be shown within the bar, because the N was only 4.  

7.     “4. Discussion”: The authors should provide appropriate discussion according to their findings.

8.     “5. Conclusion”: There was a lack of relation between adipose stem cells-derived extracellular vesicles and regulatory T cells. It is unreasonable to draw the conclusions from the findings of the study.

9.     The article would benefit from English language editing by a scientific editor who is a native English speaker.

Comments on the Quality of English Language

 The article would benefit from English language editing by a scientific editor who is a native English speaker.

Author Response

Reviewer 1:

Thank you for careful review of our manuscript.

Comment 1: “Abstract”: The authors should provide definitions for any abbreviated terms at first use. For terms used once or twice, abbreviation may not be needed. In addition, the findings from the study were not solid enough to draw the conclusions.

Response 1: We provided definitions for abbreviated terms in the abstract as your suggestion. As you pointed out, we added the missing findings from the study.

Comment 2: “2.3. Intranasal administration of SCGB1C1”: Is there any relations between SCGB1C1 and adipose stem cells-derived extracellular vesicles? If no, the title of the manuscript may not be appropriate.

Response 2: Recent our study reported that gene expression levels of SCGB1C1 decreased in lung tissue of asthmatic mice, but that SCGB1C1 expression significantly increased after treatment with ASC-derived EVs. (Kim SD, Kang SA, Kim YW, Yu HS, Cho KS, Roh HJ. Screening and functional pathway analysis of pulmonary genes associated with suppression of allergic airway inflammation by adipose stem cell-derived extracellular vesicles. Stem Cells Int. 2020;2020:5684250). These results suggested that SCGB1C1 may play a critical role in the immune suppression mechanisms of ASC-derived EVs in asthmatic mice. However, the exact role of SCGB1C1 in the suppression of allergic airway inflammation by ASC-derived EVs remains to be elucidated. Therefore, we performed this study to evaluate the immunomodulatory effect of SCGB1C1 in asthmatic mice. As you pointed out, we changed the title of this manuscript.     

Comment 3:  Figure 1: It is important to mention the number of mice for each group.

Response 3: Four mice were used in each group. We added this in the methods and figure legends.

Comment 4: In Figure 1A, “Sensitization”, “Challenge”, and “Treatment” may be more appropriate to illustrate the time points of intervention. The agents used for sensitization, challenge, and treatment can be describe as the footers of the table in Figure 1B.

Response 4: As you suggested, we changed the Figure 1A. The agents used for sensitization, challenge, and treatment were described in the figure legends.

Comment 5: “2.5. Differential cell counting in bronchoalveolar lavage fluid (BALF)”: A brief description of the procedure to collect BLAF is needed. The flush volume, the collection volume, and the times of flushes have great impact on the results.

Response 5: As you pointed out, we added the flush volume, the collection volume, and the times of flushes in the methods. 

Comment 6: In all figures of the Results, the number of mice for each group should be addressed. The author mentioned that “Data are expressed as the mean ± SD of four independent experiments, each performed in triplicate”. For a bar plot, the average datum of an individual can be shown within the bar, because the N was only 4.  

Response 6: Four mice were for all experimental groups. Differential cell counting in BALF, cytokine measurement by ELISA, and FACS experiments were performed with four samples per group. Histologic analysis was performed with four mice per group at two locations per mouse, resulting in eight data per group. Furthermore, samples from four independent experiments were derived from four individuals.  

Comment 7: “4. Discussion”: The authors should provide appropriate discussion according to their findings.

Response 7: As you pointed out, we modified the discussion according to our findings. 

Comment 8: “5. Conclusion”: There was a lack of relation between adipose stem cells-derived extracellular vesicles and regulatory T cells. It is unreasonable to draw the conclusions from the findings of the study.

Response 8: In the recent our study, intranasal administration of adipose stem cells (ASCs)-derived extracellular vesicles (EVs) significantly increased the populations of CD4+CD25+Foxp3+ regulatory T cells (Tregs) in asthmatic mice compared to the OVA group. ASCs-derived EVs improved airway hyperresponsiveness and allergic airway inflammation through the induction of Tregs expansion. (Mun SJ, Kang SA, Park HK, Yu HS, Cho KS, Roh HJ. Intranasally administrated extracellular vesicles from adipose stem cells have immunomodulatory effects in a mouse model of asthma. Stem Cells Int. 2021;2021:6686625). Therefore, the relation between adipose stem cells-derived extracellular vesicles and regulatory T cells was confirmed by the previous our study.

Comment 9: The article would benefit from English language editing by a scientific editor who is a native English speaker.

Response 9: Because of the urgency of the resubmission deadline, we chose a faster review process instead of comprehensive full-text English language editing. Therefore, this manuscript was edited by a native English-speaking editor affiliated with the author’s institution.

Reviewer 2 Report

Comments and Suggestions for Authors

Dear authors,

The manuscript is well written and structured, however some questions need to be answered before publcation:

What are extracellular vesicles (EVs), and how are they related to the study?

Why is the gene SCGB1C1 significant in the context of allergic airway inflammation in asthmatic mice?

Describe the method used to sensitize and challenge the C57BL/6 mice with OVA in this study.

How was SCGB1C1 administered to the mice, and what was the dosage?

What parameters were evaluated to assess the effect of SCGB1C1 on allergic airway inflammation?

What were the key findings regarding AHR, eosinophils in BALF, and lung inflammation after SCGB1C1 treatment?

How did SCGB1C1 affect the cytokine profiles in the BALF and LLN?

What impact did SCGB1C1 have on the populations of CD4+CD25+Foxp3+ T cells in asthmatic mice?

Based on the results, what role might SCGB1C1 play in the treatment of allergic airway inflammation?

Author Response

Reviewer 2:

Thank you for careful review of our manuscript.

The manuscript is well written and structured, however some questions need to be answered before publication:

Comment 1: What are extracellular vesicles (EVs), and how are they related to the study?

Response 1: EVs are spherical bi-layered proteolipid secreted from almost all live cells into the extracellular spaces. Depending on origin, size, and biogenesis, EVs can be subdivided as apoptotic bodies, microvesicles, and exosomes. MSC-derived EVs, which are the main paracrine effector of stem cells, play a crucial role in intracellular communication by transferring important biomolecules. The MSCs-derived EVs were as effective as MSCs themselves in improving allergic airway diseases. The immunomodulatory effects of MSCs-derived EVs in allergic airway inflammation may be mediated by the upregulation of regulatory T cells and increased expression of SCGB1C1. However, the role of SCGB1C1 in the suppression of allergic airway inflammation by MSC-derived EVs remains to be elucidated. Therefore, we performed this study to evaluate the immunomodulatory effect of SCGB1C1 in asthmatic mice. We added in the discussion.    

Comment 2: Why is the gene SCGB1C1 significant in the context of allergic airway inflammation in asthmatic mice?

Response 2: Recent our study reported that gene expression levels of SCGB1C1 decreased in lung tissue of asthmatic mice, but that SCGB1C1 expression significantly increased after treatment with ASC-derived EVs. (Kim SD, Kang SA, Kim YW, Yu HS, Cho KS, Roh HJ. Screening and functional pathway analysis of pulmonary genes associated with suppression of allergic airway inflammation by adipose stem cell-derived extracellular vesicles. Stem Cells Int. 2020;2020:5684250). These results suggested that SCGB1C1 may play a critical role in the immune suppression mechanisms of ASC-derived EVs in asthmatic mice. However, the exact role of SCGB1C1 in the suppression of allergic airway inflammation by ASC-derived EVs remains to be elucidated. Therefore, we performed this study to evaluate the immunomodulatory effect of SCGB1C1 in asthmatic mice.    

Comment 3: Describe the method used to sensitize and challenge the C57BL/6 mice with OVA in this study.

Response 3: As described in the methods, C57BL/6 mice were sensitized by intraperitoneal injection with 75 μg of OVA and 2 mg of aluminum hydroxide in 200 μL of PBS on days 0, 1, 7, and 8. Subsequently, the mice were challenged intranasally with 50 μg of OVA in 50 μL PBS on days 14, 15, 21, and 22 after the initial sensitization. 

Comment 4: How was SCGB1C1 administered to the mice, and what was the dosage?

Response 4: 5 μg/50 μl of SCGB1C1 were administrated intranasally before OVA challenge in an OVA induced asthmatic mouse model.

Comment 5: What parameters were evaluated to assess the effect of SCGB1C1 on allergic airway inflammation?

Response 5: To assess the effect of SCGB1C1 on allergic airway inflammation, we evaluated airway hyperresponsiveness, total inflammatory cells and eosinophils in the bronchoalveolar lavage fluid (BALF), lung histology, serum immunoglobulin, cytokine profiles of BALF and lung draining lymph nodes (LLN), and T cell populations in LLNs.    

Comment 6: What were the key findings regarding AHR, eosinophils in BALF, and lung inflammation after SCGB1C1 treatment?

Response 6: SCGB1C1 treatment significantly decreased AHR and the numbers eosinophils in the BALF of asthmatic mice. Furthermore, SCGB1C1 treatment induced a significant reduction in the infiltration of inflammatory cells and goblet cells hyperplasia in asthmatic mice.

Comment 7: How did SCGB1C1 affect the cytokine profiles in the BALF and LLN?

Response 7: Intranasal SCGB1C1 treatment significantly decreased IL-5 in the BALF and IL-4 in the LLN of asthmatic mice. In contrast, intranasal SCGB1C1 treatment remarkably increased IL-10 and TGF-β in the BALF of asthmatic mice.

Comment 8: What impact did SCGB1C1 have on the populations of CD4+CD25+Foxp3+ T cells in asthmatic mice?

Response 8: The populations of CD4+CD25+Foxp3+ T cells in LLN of asthmatic mice were markedly increased by intranasal administration of SCGB1C1.

Comment 9: Based on the results, what role might SCGB1C1 play in the treatment of allergic airway inflammation?

Response 9: SCGB1C1 treatment significantly reduced allergic airway inflammation and improving lung function through the induction of Tregs expansion in asthmatic mice. Therefore, SCGB1C1 play a critical role in suppression of allergic airway inflammation by ASC-derived EVs.

Reviewer 3 Report

Comments and Suggestions for Authors

Dear Authors,

The manuscript entitled "SCGB1C1 play a critical role in suppression of allergic airway inflammation through adipose stem cells-derived extracellular vesicles" is a very well prepared article and indeed have a great interest for the authors. Only minor revisions, i have recommended for the submitted manuscript.

1) The authors in the introduction section support that the SCGB1C1 can be found in exosomes derived from adipose stem cells, however, this peptide was chemically synthesized and not derived fromt the exosomes. For this purpose the title of the manuscript must be changed to avoid any confusion for the readers.

2) Experimental approaches and statistical analysis have been well performed, no change is required.

3) In the discussion section, i think that the first paragraph should be rewritten, due to the fact that the authors didn't use SCGB1C1 from exosomes derived from adipose stem cells.

Author Response

Reviewer 3:

Thank you for careful review of our manuscript.

The manuscript entitled "SCGB1C1 play a critical role in suppression of allergic airway inflammation through adipose stem cells-derived extracellular vesicles" is a very well prepared article and indeed have a great interest for the authors. Only minor revisions, I have recommended for the submitted manuscript.

Comment 1: The authors in the introduction section support that the SCGB1C1 can be found in exosomes derived from adipose stem cells, however, this peptide was chemically synthesized and not derived from the exosomes. For this purpose the title of the manuscript must be changed to avoid any confusion for the readers.

Response 1: As you pointed out, we changed the title of this manuscript.

Comment 2: Experimental approaches and statistical analysis have been well performed, no change is required.

Response 2: We modified some sentence in the materials and methods sections according to the other reviewers’ comment.

Comment 3: In the discussion section, I think that the first paragraph should be rewritten, due to the fact that the authors didn't use SCGB1C1 from exosomes derived from adipose stem cells.

Response 3: As you pointed out, we did not use SCGB1C1 from exosomes derived from adipose stem cells. However, recent our study reported that gene expression levels of SCGB1C1 decreased in lung tissue of asthmatic mice, but that SCGB1C1 expression significantly increased after treatment with ASC-derived EVs. (Kim SD, Kang SA, Kim YW, Yu HS, Cho KS, Roh HJ. Screening and functional pathway analysis of pulmonary genes associated with suppression of allergic airway inflammation by adipose stem cell-derived extracellular vesicles. Stem Cells Int. 2020;2020:5684250). These results suggested that identified SCGB1C1 protein may play a critical role in the immune suppression mechanisms of ASC-derived EVs in asthmatic mice. However, the exact role of SCGB1C1 in the suppression of allergic airway inflammation by ASC-derived EVs remains to be elucidated. Therefore, we performed this study to evaluate the immunomodulatory effect of SCGB1C1 in asthmatic mice. As you pointed out, we changed the first paragraph in the discussion section.    

Round 2

Reviewer 1 Report

Comments and Suggestions for Authors

I don't have additional comment.

Comments on the Quality of English Language

I don't have additional comment.